# LoR-VP: Low-Rank Visual Prompting for Efficient Vision Model Adaptation

**Can Jin**[1], **Ying Li**[2], **Mingyu Zhao**[1], **Shiyu Zhao**[1], **Zhenting Wang**[1], **Xiaoxiao He**[1],
**Ligong Han**[3,4], **Tong Che**[5], **Dimitris N. Metaxas**[1]
[1]Rutgers University, [2]Zhejiang University, [3]Red Hat AI Innovation,
[4]MIT-IBM Watson AI Lab, [5]NVIDIA Research

## ABSTRACT

Visual prompting has gained popularity as a method for adapting pre-trained models to specific tasks, particularly in the realm of parameter-efficient tuning. However, existing visual prompting techniques often pad the prompt parameters around the image, limiting the interaction between the visual prompts and the original image to a small set of patches while neglecting the inductive bias present in shared information across different patches. In this study, we conduct a thorough preliminary investigation to identify and address these limitations. We propose a novel visual prompt design, introducing **Lo**w-**R**ank matrix multiplication for **V**isual **P**rompting (LoR-VP), which enables shared and patch-specific information across rows and columns of image pixels. Extensive experiments across seven network architectures and four datasets demonstrate significant improvements in both performance and efficiency compared to state-of-the-art visual prompting methods, achieving up to $6\times$ faster training times, utilizing $18\times$ fewer visual prompt parameters, and delivering a $3.1\%$ improvement in performance. Code is available at `https://github.com/jincan333/LoR-VP`.

## 1 INTRODUCTION

Many applications in computer vision (CV) and natural language processing (NLP) rely on adapting large-scale, pre-trained models to multiple downstream tasks (Liu et al., 2021; Dosovitskiy et al., 2020; Ridnik et al., 2021; Brown et al., 2020; Peng et al., 2024). Recent advances in large language models (LLMs) have highlighted data-centric techniques such as in-context learning (Brown et al., 2020; Shin et al., 2020; Liu et al., 2022) and prompting (Li & Liang, 2021; Liu et al., 2023). These techniques well-designed prompts or input templates to significantly enhance the performance of LLMs across a wide range of tasks. Inspired by these methods, visual prompting has gained substantial attention as a means of adapting pre-trained vision models by modifying input pixels or output transformations (Bahng et al., 2022; Chen et al., 2023; Tsao et al., 2024).

Existing visual prompting methods, such as those proposed by CLIP-VP (Bahng et al., 2022), ILM-VP (Chen et al., 2023), and AutoVP (Tsao et al., 2024), have demonstrated the capability to enhance the performance of pre-trained vision models across various downstream tasks. A natural question arises: why does the addition or padding of tunable parameters to the original image pixels improves adaptation performance? A plausible explanation is that the introduced visual prompts (VPs) provide task-specific information that not only alters the representation of the original images but also influences the attention distribution across image patches. This is particularly significant in pre-trained models such as Vision Transformers (ViTs) (Dosovitskiy et al., 2020), where VPs interact with patch tokens and guide the model's attention to different parts of the image.

However, current VP designs primarily focus on adding or padding tunable parameters at the periphery of the image (see Part 1 of Figure 1 for existing VP designs), thus only allowing boundary patches to be modified, while the central patches remain unchanged. These designs present two notable limitations: (1) The VP parameters are restricted to interacting with the original image in a limited set of patches, leaving a substantial portion of the image unmodified. As a result, VPs

---

Correspondence to: Can Jin `<can.jin@rutgers.edu>`

can only influence the model's interpretation of specific regions of the image, while other regions-potentially containing critical information-remain unaffected. (2) The VPs applied to each patch operate independently, disregarding the inductive biases present in the shared information and positional encoding across different patches. For instance, one patch may represent part of an object, while an adjacent patch represents another part of the same object. By ignoring these inter-patch relationships, visual prompting may limit the model's ability to capture the global context of the image effectively.

To address these limitations, we first conduct a thorough investigation to understand the shortcomings of existing methods. Building on this analysis, we propose a novel visual prompting method, termed LoR-VP, aimed at more efficient and effective adaptation of vision models. Our approach not only influences every patch of the image but also introduces inductive biases between the rows and columns of image patches. By leveraging LoR-VP, we achieve superior performance compared to state-of-the-art (SOTA) methods while significantly reducing the number of parameters required. In summary, our contributions are organized around the following three thrusts:

* (Preliminary Study) We conduct an in-depth preliminary study to identify and illustrate the limitations present in current visual prompting methods and explore potential solutions to overcome these challenges.

* (Novel Approach) To mitigate these limitations, we propose a novel visual prompting technique, named LoR-VP, which optimizes the visual prompts uniformly across all patches and introduces inductive biases between the rows and columns of the image patches.

* (Experiments) We perform extensive experiments across a wide range of large-scale models and datasets. The empirical results consistently demonstrate the significant improvements in both performance and efficiency achieved by LoR-VP, validating its practical effectiveness. For instance, LoR-VP surpasses the previous SOTA method, AutoVP (Tsao et al., 2024), by an average of $3.1\%$ on seven network architectures and four datasets using $6\times$ less training time.

## 2 RELATED WORKS

### 2.1 VISUAL PROMPTING

The concept of prompting originated in the field of NLP as a technique for adapting pre-trained models to downstream tasks (Shin et al., 2020; Liu et al., 2022; Li & Liang, 2021; Liu et al., 2023; Han et al., 2023; 2024; Jin et al., 2023). This design philosophy was later extended to CV by Bahng et al. (2022), who introduced tunable parameters directly into input images to create what is known as a Visual Prompt (VP). A typical VP framework consists of two primary modules: input design and output transformation (Bahng et al., 2022; Tsai et al., 2020; Tsao et al., 2024; Cai et al., 2024). Various strategies have been proposed for constructing VPs. For instance, Bahng et al. (2022) modify input images by adding a frame of visual prompting parameters, whereas Chen et al. (2023) incorporate the visual prompting parameters around resized images. Wu et al. (2022) explore efficient methods for generating visual prompts that enhance performance across different tasks, and Oh et al. (2023) develop visual prompts designed for adapting models to black-box, inaccessible models. Since the output logits of pre-trained models remain in the source domain, an additional output transformation (e.g., label mapping) is required to accurately predict the targets. A simple approach is to randomly map source labels (RLM) onto target labels. Tsai et al. (2020) propose a frequency-based label mapping (FLM) technique, which derives the mapping based on frequency statistics. Chen et al. (2023) further introduces iterative label mapping (ILM), which improves the performance of visual prompting. Yang et al. (2023) proposes a semantics-based label mapping approach that aligns source and target classes based on semantic similarity. Additionally, Tsao et al. (2024) introduces full mapping (FM), utilizing an automated system to select the most appropriate label mapping (LM) method to optimize performance across diverse downstream tasks.

### 2.2 LOW-RANK STRUCTURES IN DEEP LEARNING

Low-rank structures are widely observed in machine learning, as many problems inherently exhibit low-rank properties (Li et al., 2016; Cai et al., 2010; Li et al., 2018; Grasedyck et al., 2013).

It has been found that for numerous deep learning tasks, especially those involving heavily over-parameterized neural networks, the resulting models tend to exhibit low-rank characteristics after training (Oymak et al., 2019; Khodak et al., 2021). Some prior work has explicitly integrated low-rank constraints during the training process of neural networks (Sainath et al., 2013; Zhang et al., 2014; Zhao et al., 2016). From a theoretical perspective, neural networks have been shown to outperform classical learning methods, including finite-width neural tangent kernels, when the underlying concept class has a low-rank structure (Allen-Zhu et al., 2019; Li & Liang, 2018; Ghorbani et al., 2020; Allen-Zhu & Li, 2019; 2020). Additionally, Allen-Zhu et al. (2020) highlight that low-rank adaptations can be beneficial in adversarial training scenarios. The LoRA method, introduced by Hu et al. (2021), along with its variants (Zhang, 2023; Yeh, 2023), is particularly noteworthy for not introducing additional inference burdens, thus improving the parameter efficiency of adapting large pre-trained models. These methods employ low-rank matrices to approximate weight updates during fine-tuning, enabling a seamless integration with pre-trained weights prior to inference.

## 3 PRELIMINARY STUDY

### 3.1 ANALYSIS OF PAD PROMPTING

Visual prompting is proposed to address the problem of adapting a pre-trained source model to downstream tasks without fine-tuning the network weights. Consider a downstream target image dataset $\mathcal{D} = \{(\mathbf{x}_1, y_1), ..., (\mathbf{x}_n, y_n)\}$ with typical color channels $c$ (usually 3) and a pre-trained vision model $f$ with a resolution of $L \times L$ (the default value of $L$ is 224 for simplicity). Visual prompting modifies the images by adding tunable parameters to the image pixels. Among various methods, Pad Prompting(Tsao et al., 2024; Chen et al., 2023; Bahng et al., 2022) is the most prevalent, which involves resizing the initial images to a specific size $s$ (typically smaller than $L$, such as 128), and then surrounding the resized image with a tunable parameter border of size $p$ such that $s + 2p = L$, resulting in a prompted image of dimensions $L \times L$. An illustration of Pad Prompting is depicted in Part 1 of Figure 1. The optimal values for $s$ and $p$ generally vary across different models and tasks. AutoVP, currently the SOTA in visual prompting, automates the selection of $s$ and $p$ to enhance performance across various models and tasks (Tsao et al., 2024).

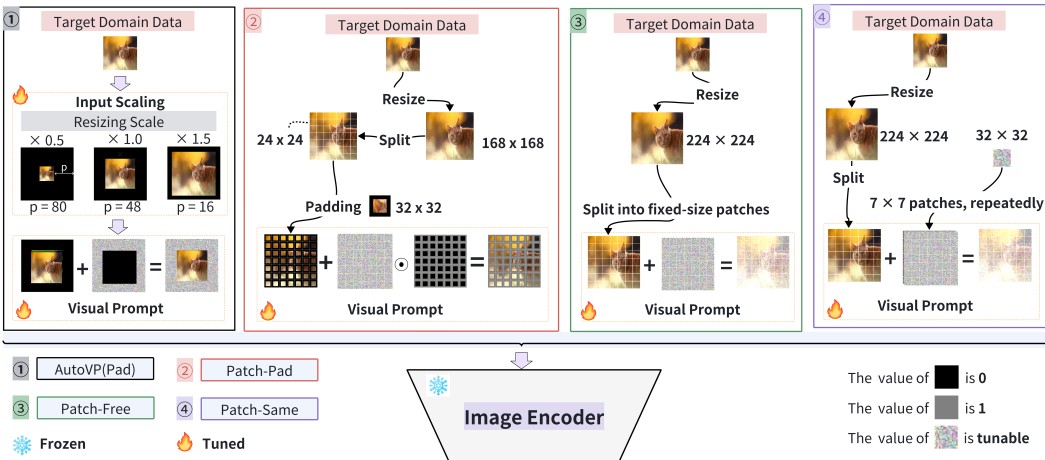

Figure 1: Illustration of various visual prompting methods applied to target domain data: ❶ **AutoVP(Pad):** Focuses on optimizing the balance between image scaling and tunable parameter integration to enhance model responsiveness. ❷ **Patch-Pad:** Aims to enhance localized learning by surrounding each image patch with tunable visual prompts. ❸ **Patch-Free:** Provides maximum adaptability by allowing independent tuning of visual prompts for each patch, catering to diverse feature requirements across the image. ❹ **Patch-Same:** Promotes consistency in model training by applying uniform visual prompts across all patches, ensuring coherent feature learning across input.

Despite AutoVP's automated process for optimizing prompt size, Pad Prompting inherently only adjusts the prompts in the peripheral patches of the resized images, leaving the central part unchanged.

Furthermore, the parameters in different patches are optimized independently, disregarding the inductive biases that arise from shared information and positional encoding across patches. We hypothesize that a more effective visual prompting strategy would allow interaction with each patch while also considering the inductive biases among them. This hypothesis has led us to explore new VP designs:

- **Patch-Pad**: As shown in Part 2 of Figure 1, after resizing the image, we evenly split it into different patches and then pad the tunable parameters around each patch to form a resolution $L \times L$ image. For instance, when using an ImageNet-21K (Deng et al., 2009) pre-trained ViT-B/32 (Dosovitskiy et al., 2020) and fine-tuned on ImageNet-1K model, the resize image is split into $7 \times 7$ patches, and each patch is padded to a size of $32 \times 32$ using tunable parameters. The Patch-Pad method can influence each patch of the image but split the image into discontiguous parts and potentially devastate the information in the images.

- **Patch-Free**: As shown in Part 3 of Figure 1, to avoid dropping information in the images, we directly resize the image to resolution $L \times L$, then evenly add independent tunable parameters (initialized as 0) to each patch of the resized image. The Patch-Free design can influence each patch of the image without splitting the resized image thus maintaining the information of the image, but this design updates the patch VPs independently and doesn't consider shared information on different patches.

- **Patch-Same**: As shown in Part 4 of Figure 1, to enable shared prompting among each patch, we initialize a patch of tunable parameters and repeatedly add it to all patches of the image. Patch-Same enables shared visual prompting among different patches but constrains the shared information as the same for all patches.

## 3.2 PERFORMANCE INVESTIGATION OF DIFFERENT VP DESIGNS

To investigate the performance of the aforementioned four VP designs, we conducted experiments using ImageNet-21K pre-trained ViT-B/32 and ViT-B/16 (both fine-tuned on ImageNet-1K) on CIFAR10/100 (Krizhevsky et al., 2009). The performance of AutoVP (Pad Prompting) was used as a benchmark, with all methods employing a fully connected label (FM) (Tsao et al., 2024) for fair comparison. The results, depicted in Figure 2, indicate that ❶ Patch-Pad underperforms across all models and datasets, the most likely reason is it split the image patches thus might damage the image information. ❷ Patch-Free outperforms Patch-Pad, confirming that maintaining image continuity is beneficial. However, Patch-Free is less effective than Patch-Same, which suggests that shared visual prompting can enhance performance. ❸ Patch-Same outperforms AutoVP, underscoring the importance of shared prompting information across patches. ❹ The performance gap between Patch-Same and Pad Prompting shrinks in models that have smaller patch sizes, suggesting that when using the ViT-B/16 model, Patch-Same constrains more patches to learn a smaller visual prompt, which may bring too strong constraints to the visual prompts, and a better way is utilizing visual prompts that not only introduce inductive bias in different patches but also allow for patch-specific visual prompting.

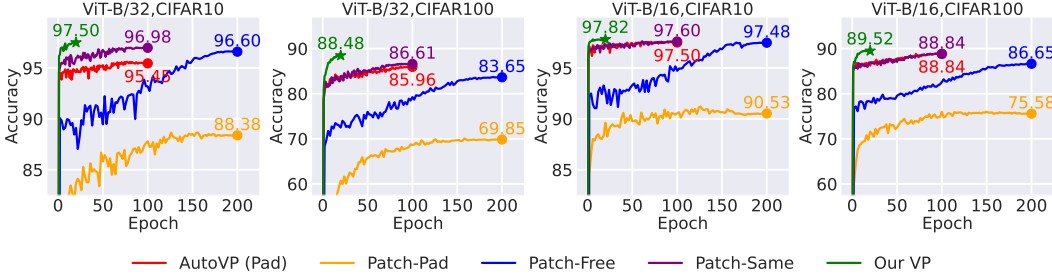

Figure 2: **Preliminary Investigation Results.** Performance comparison of various VP designs. Our VP method demonstrates competitive or superior performance in several configurations. The final performance of each method is marked by ★ or •, with all results averaged over three runs.

## 4 METHODOLOGY

Inspired by the observations in Section 3, we propose a novel visual prompt design that facilitates prompting across all patches while enabling both shared and patch-specific information. This approach leverages low-rank matrix multiplication to efficiently manage visual prompts.

### 4.1 LOW-RANK VISUAL PROMPTING

In order to facilitate comprehensive pixel information and interaction across all patches, we resize the image to a uniform size of $L \times L$. This resizing strategy is designed to minimize information loss, which is a common issue with the traditional Pad Prompting method that often resizes images to dimensions smaller than $L$. To enable the sharing of visual prompt information across different patches, we introduce two low-rank matrix parameters, $\mathbf{B} \in \mathbb{R}^{c \times L \times r}$ and $\mathbf{A} \in \mathbb{R}^{c \times r \times L}$, where $r \ll L$. The product of these matrices, $\mathbf{B} \cdot \mathbf{A}$, serves as the visual prompt.

This configuration allows the visual prompt to act as a linear combination of the row vectors in $\mathbf{A}$ and the column vectors in $\mathbf{B}$, facilitating shared information across the rows and columns of the image. Additionally, this design supports patch-specific information, as the coefficients of each row and column are independently adjustable. Based on our observations in Section 3, this approach to visual prompting is likely to yield superior performance. The visual prompt is directly added to the resized image, resulting in the prompted image being expressed as:

$$\mathcal{P}(\mathbf{x}) = \text{Resize}_L(\mathbf{x}) + \mathbf{B} \cdot \mathbf{A}, \quad \mathbf{x} \in \mathcal{D}, \tag{1}$$

where $\text{Resize}_L(\cdot)$ resizes the image $\mathbf{x}$ into a size of $L \times L$, and matrices $\mathbf{B}$ and $\mathbf{A}$ are the initialized visual prompt parameters, we utilize zero initialization of $\mathbf{B}$ and a random Gaussian initialization of $\mathbf{A}$ so $\mathbf{B} \cdot \mathbf{A}$ is zero at the beginning of training.

Utilizing a rank $r = 4$ in $\mathbf{B}$ and $\mathbf{A}$, we conduct experiments using the same configurations as Section 3. From the experimental results shown in Figure 2, we can observe that our **Lo**w-**R**ank matrices multiplication **V**isual **P**rompting (**LoR-VP** ) achieve the best performance among all designs, further validate our hypothesis in Section 3.

This method simplifies the visual prompting process compared to Pad Prompting, which involves complex resizing, padding, and mask manipulations. By employing low-rank matrices, we reduce the number of tunable parameters from $cL^2$ to $crL$, enhancing parameter efficiency significantly.

### 4.2 OUTPUT TRANSFORMATION

The output of the pre-trained model $f$ on the prompted image $\mathcal{P}(\mathbf{x})$ remains in the source domain. To align these predictions with target labels in downstream tasks, we apply an output transformation, denoted as $\mathcal{M}$:

$$\underset{\boldsymbol{\delta}, \mathcal{M}}{\text{minimize}} \quad \mathbb{E}_{(\mathbf{x}, y) \in \mathcal{D}} \mathcal{L}(\mathcal{M}(f(\mathcal{P}(\mathbf{x}))), y), \tag{2}$$

We utilize Linear Probing (LP) as our output transformation method, which adjusts the output features of the classifier head to suit downstream classes. This method proves more efficient than existing methods such as iterative label mapping (ILM) (Chen et al., 2023) and fully connected layer mapping (FM) (Tsao et al., 2024) on large models and datasets. For instance, when using ImageNet-21K pre-trained Swin-B (Liu et al., 2021) and tuning on ImageNet-1K, ILM costs too much time and GPU storage to calculate and store the mapping sequences (a $21,841 \times 1,000$ matrix) and AutoVP also achieves inferior performance due to the ineffectiveness to learn a $21,841 \times 1,000$ full connected layer (see Figure 5 for performance details).

## 5 EXPERIMENTS

To assess the effectiveness and efficiency of our novel visual prompting method, we adopt the most widely used evaluation protocol for VPs, i.e., evaluating models pre-trained on large datasets across

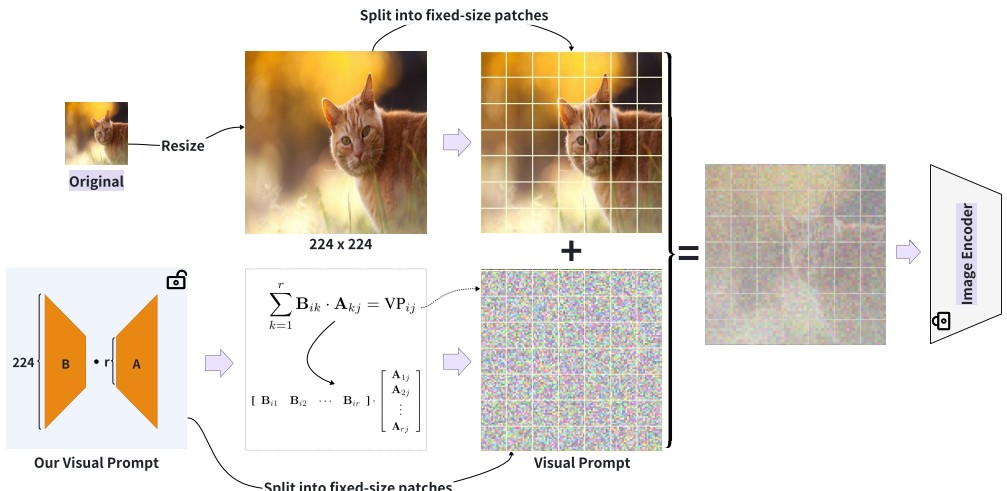

Figure 3: **Our VP Design.** We resize the image to a resolution of $L \times L$ and initialize two low-rank matrices $\mathbf{B}$ and $\mathbf{A}$ as tunable parameters. $\mathbf{B} \cdot \mathbf{A}$ serves as the visual prompt and is directly added to the resized images. This design allows shared information in rows and columns and also allows patch-specific information in different patches.

various visual domains. Unlike previous works such as ILM-VP (Chen et al., 2023) and AutoVP (Tsao et al., 2024), which primarily utilize ImageNet-1K (Deng et al., 2009) pre-trained models and fine-tune on smaller downstream datasets, we extend our exploration to larger pre-training datasets, such as ImageNet-21K (Deng et al., 2009), as well as larger downstream datasets, including ImageNet-1K, to examine the scalability of existing VP methods. Furthermore, we conduct extensive empirical evaluations, focusing on the following aspects: (1) Demonstrating the superior performance and faster convergence of LoR-VP across different datasets and architectures; (2) Investigating the out-of-distribution robustness of LoR-VP; (3) Showcasing the efficiency of LoR-VP in terms of training epochs, runtime, and parameter usage, etc; (4) Performing ablation studies to evaluate the effectiveness of our VP approach under various label mapping methods, the optimal rank configuration in LoR-VP, and the contribution of different components within LoR-VP.

## 5.1 IMPLEMENTATION DETAILS

**Datasets.** For pre-training, we utilize the ImageNet-1K dataset (Deng et al., 2009), which contains 1K classes and 1.3M images, the ImageNet-21K-P dataset (Ridnik et al., 2021), comprising 11K classes and 12M images, and the ImageNet-21K dataset (Deng et al., 2009), which includes 21K classes and 14M images. We evaluate the effectiveness and efficiency of LoR-VP across four downstream datasets: ImageNet-1K, Tiny-ImageNet (Le & Yang, 2015), and CIFAR-10/100 (Krizhevsky et al., 2009). To assess the out-of-distribution robustness of LoR-VP, we conduct experiments on ImageNet-R (Hendrycks et al., 2021a), ImageNet-Sketch (Wang et al., 2019), ImageNet-A (Hendrycks et al., 2021b), and ImageNet-V2 (Recht et al., 2019). Additional details about the datasets are in Table 6.

**Networks.** We employ six architectures for our experiments, all of which operate at a resolution of $224 \times 224$. (1) ResNet-18 and ResNet-50 (He et al., 2016) pre-trained on ImageNet-1K, and ViT-B/32 (Dosovitskiy et al., 2020) pre-trained on ImageNet-21K and fine-tuned on ImageNet-1K, each with a classifier head of 1000 classes; (2) ResNet-50-P and ViT-B/16-P (Dosovitskiy et al., 2020), pre-trained on ImageNet-21K-P, with classifier heads for 11,221 classes; (3) Swin-B (Liu et al., 2021), pre-trained on ImageNet-21K, with a classifier head for 21,841 classes; (4) CLIP (Radford et al., 2021), a vision-language model that uses a ViT-B/32 architecture as its vision encoder. The

weights for these models are all publicly available through the official PyTorch Model Zoo or the Hugging Face Timm Library. Further details of the network architectures can be found in Table 7.

**Baselines.** We select four representative SOTA methods as our baselines: (1) *CLIP-VP* (Bahng et al., 2022), which extends prompt tuning to the computer vision domain by incorporating prompt parameters directly into input images using CLIP models; (2) *ILM-VP* (Chen et al., 2023), which explores the impact of frequency-based label mapping (FLM) in visual prompting and introduces iterative label mapping (ILM) for improved performance; (3) *AutoVP* (Tsao et al., 2024), a SOTA method in visual prompting that automates the selection of VP configurations, including prompt sizes and label mapping (LM) strategies—our experiments use the optimal configuration provided by AutoVP; (4) *LP*, which modifies the classifier head of the pre-trained model to adapt to downstream tasks, a commonly used technique in transfer learning, serving as a baseline akin to LoR-VP without the novel VPs introduced in our approach.

**Training and Evaluation.** The results for the baseline methods, including CLIP-VP, ILM-VP, and AutoVP, are reproduced using the same configurations as described in their respective original papers. For LoR-VP, we resize all input images to $224 \times 224$ and use a rank of 4 in our VP design. As a result, the two sets of parameters in LoR-VP have dimensions of $3 \times 224 \times 4$ and $3 \times 4 \times 224$, respectively, meaning that the total number of parameters in the visual prompts is only 5K. The optimal hyperparameters for LoR-VP are determined through grid search. All experiments are conducted on NVIDIA Quadro RTX8000 GPUs with 48GB of memory. Additional implementation details for LoR-VP are provided in Table 8.

## 5.2 MAIN RESULTS

**Performance of ImageNet-1K and CLIP Pre-trained Models.** To demonstrate the effectiveness of LoR-VP on widely used ImageNet-1K pre-trained models, we evaluate its performance across several downstream datasets using ImageNet-21K pre-trained ViT-B/32 (fine-tuned on ImageNet-1K), ImageNet-1K pre-trained ResNet-18 and ResNet-50, as well as CLIP models. As shown in Figure 4, we can observe that: ❶ LoR-VP consistently outperforms all baselines across all network and dataset combinations, achieving an average improvement of 3.2% and 2.1% over AutoVP and LP, respectively. ❷ LoR-VP converges significantly faster than the baselines, reaching optimal performance with $5\times$ fewer training epochs than AutoVP and $10\times$ fewer than ILM-VP.

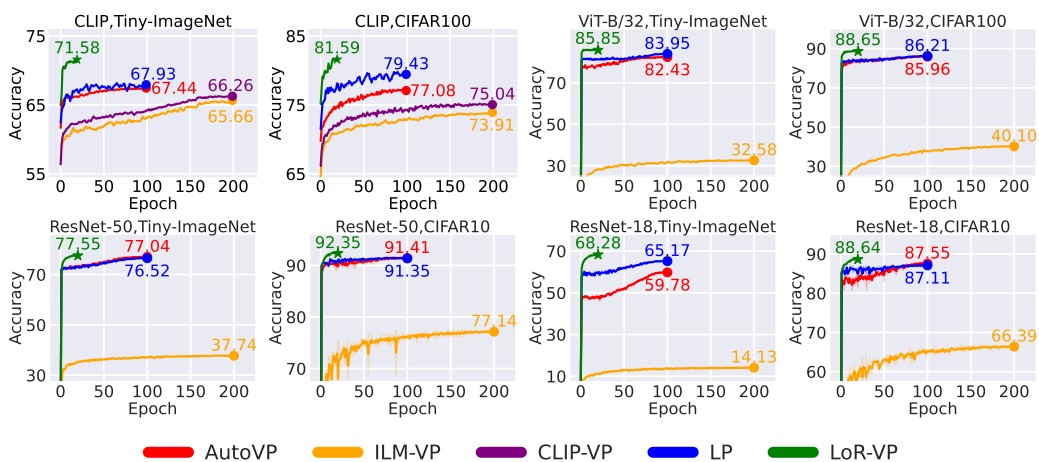

Figure 4: **Performance of ImageNet-1K and CLIP Pre-trained Models on Downstream Datasets.** Overview of the performance of LoR-VP compared to four baseline methods. The final performance of each method is indicated by ★ or •, and all results are averaged over three runs. LoR-VP consistently outperforms all baselines across various models and datasets.

---

https://pytorch.org/vision/stable/models.html
https://huggingface.co/models?library=timm

**Performance of ImageNet-21K Pre-trained Models.** To further evaluate the effectiveness of LOR-VP and existing visual prompting methods on larger models and datasets with a greater number of classifier classes, we conduct experiments using ImageNet-21K-P pre-trained ResNet-50-P and ViT-B/16-P, as well as ImageNet-21K pre-trained Swin-B models, tuning them on ImageNet-1K and Tiny-ImageNet. These models have significantly more classifier output features compared to ImageNet-1K pre-trained models, providing additional evidence of the effectiveness of LOR-VP and other VP methods on large-scale models and datasets. For the ImageNet-1K experiments, we focus on the strongest baselines, such as AutoVP and LP, running them for 30 epochs in line with the implementation in Liu et al. (2021), due to resource constraints. We found it challenging to run ILM-VP on our GPUs, as the ILM process is computationally expensive in terms of both training time and GPU memory. The results of the experiments are presented in Figure 5, where we observe the following: ❶ LOR-VP consistently achieves the best performance across large models and datasets, outperforming all baselines. ❷ The performance gap between LOR-VP and AutoVP increases to 5.06 on ImageNet-1K. A likely explanation is that the full mapping (FM) method used in AutoVP is less effective in this scenario, as it struggles to efficiently train a fully connected layer with 21,841 input features and 1,000 output features.

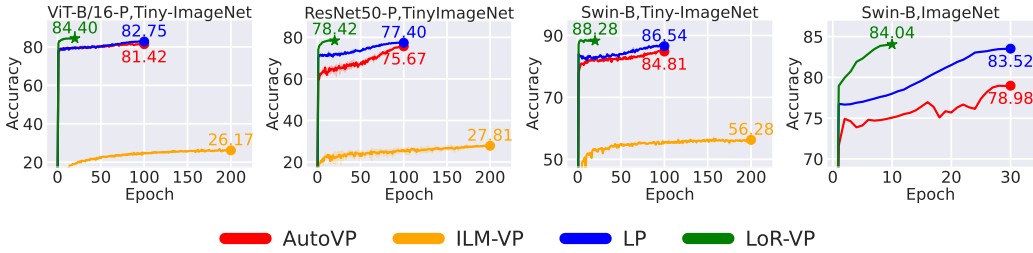

Figure 5: **Performance of ImageNet-21K Pre-trained Models on ImageNet-1K and Tiny-ImageNet.** Performance comparison of LOR-VP and four baseline methods. The models are pre-trained on either ImageNet-21K-P or ImageNet-21K and then tuned on the respective downstream datasets. The final performance results are denoted by ★ or ●. All results are averaged over three runs. LOR-VP consistently outperforms all baselines across different models and datasets.

## 5.3 ROBUSTNESS OF LOR-VP

To investigate the out-of-distribution robustness of LOR-VP and explore its potential for enhancing real-world applications, we conduct experiments using ImageNet-21K pre-trained Swin-B. We apply LOR-VP on ImageNet-1K and then evaluate the performance of the resulting model and visual prompts on four out-of-distribution datasets. The performance of LOR-VP and the two strongest baselines are presented in Table 1. We find that LOR-VP consistently demonstrates the best out-of-distribution robustness and generalization performance across all baselines, achieving an average improvement of 10.6 over AutoVP across the four datasets. These results further highlight the superior out-of-distribution robustness of LOR-VP, confirming its advantage over SOTA prompting methods in terms of generalization.

Table 1: **Out-of-Distribution Generalization Performance.** Evaluation of the out-of-distribution generalization performance using the ImageNet-21K pre-trained Swin-B, with visual prompting applied on ImageNet-1K, and tested across four out-of-distribution datasets.

| Method | Source | Target | | | |
|---|---|---|---|---|---|
| | ImageNet-1K | ImageNet-R | ImageNet-Sketch | ImageNet-A | ImageNet-V2 |
| AutoVP [ICLR24] | 78.98 | 38.14 | 28.89 | 17.91 | 67.38 |
| LP | 83.52 | 51.48 | 40.07 | 27.54 | 71.86 |
| LOR-VP | **84.04** | **52.27** | **41.13** | **27.89** | **72.38** |

## 5.4 Effieiency of LoR-VP

As shown in Figures 4 and 5, LoR-VP achieves superior performance compared to SOTA methods with fewer training epochs. To further examine the efficiency of LoR-VP in contrast to the baselines, we assess its performance using several criteria: training epochs, training time, tunable parameters (including visual prompt parameters), GPU memory usage during training, and inference latency. Evaluations are conducted using an ImageNet-21K-P pre-trained ViT-B/16-P with visual prompting applied on Tiny-ImageNet, and an ImageNet-1K pre-trained ResNet-18 with visual prompting applied on CIFAR-10. The results are presented in Table 2, we can observe that: ❶ LoR-VP converges the fastest among all methods, requiring $5\times$ fewer epochs and $6\times$ less training time compared to AutoVP, and $10\times$ fewer epochs and $15\times$ less training time compared to ILM-VP. ILM-VP, in particular, converges the slowest and incurs the highest time cost, as it requires an additional epoch for every training epoch to compute the LM sequences. ❷ LoR-VP is highly parameter-efficient. For small models and datasets, such as ResNet-18 and CIFAR-10, LoR-VP only requires 10K parameters to achieve optimal performance, which is $15\times$ and $11\times$ fewer than ILM-VP and AutoVP, respectively. Notably, LoR-VP requires just 5K visual prompt parameters, which is $18\times$ and $30\times$ fewer than AutoVP and ILM-VP, on average. ❸ GPU usage and inference speed for LoR-VP are comparable to AutoVP, whereas ILM-VP consumes the most GPU memory on larger models due to the additional computation and storage required for LM sequences. ❹ LoR-VP achieves the best performance with the fewest visual prompt parameters and the shortest training time, making it an ideal choice for adapting pre-trained vision models to downstream tasks, particularly for resource-constrained environments such as mobile devices.

Table 2: **Training and Inference Efficiency.** Comparison of the training and inference efficiency of LoR-VP, AutoVP, and ILM-VP, evaluated using ImageNet-21K-P pre-trained ViT-B/16-P on Tiny-ImageNet and ImageNet-1K pre-trained ResNet-18 on CIFAR-10.

| Network | Dataset | Method | Epochs | Time | # VP Params | # Tunable Params | GPU Usage | Latency | Accuracy |
|---|---|---|---|---|---|---|---|---|---|
| ResNet-18 | CIFAR10 | ILM-VP[CVPR23] | 200 | 5.76h | 147K | 147K | 4.51GB | 4.61ms | 66.39 |
| | | AutoVP[ICLR24] | 100 | 2.61h | 101K | 111K | 4.51GB | 4.59ms | 87.55 |
| | | LoR-VP | **20** | **0.50h** | **5K** | **10K** | **4.49GB** | **4.53ms** | **88.64** |
| ViT-B/16-P | Tiny-ImageNet | ILM-VP[CVPR23] | 200 | 25.55h | 147K | **147K** | 17.24GB | 14.55ms | 26.17 |
| | | AutoVP[ICLR24] | 100 | 8.08h | 74K | 2,318K | 13.59GB | 14.40ms | 81.42 |
| | | LoR-VP | **20** | **1.32h** | **5K** | 159K | **13.34GB** | **14.29ms** | **84.40** |

## 5.5 Albation Studies

**How does Output Transformation Impact LoR-VP's Performance?** To further explore how different output transformations affect the performance of LoR-VP, we conduct experiments by combining LoR-VP with FLM, ILM, and FM, referred to as LoR-VP w. FLM, LoR-VP w. ILM, and LoR-VP w. FM, respectively. These experiments are performed using ImageNet-21K pre-trained Swin-B, ImageNet-21K-P pre-trained ViT-B/16-P, ImageNet-21K pre-trained ViT-B/32 (fine-tuned on ImageNet-1K), and ImageNet-1K pre-trained ResNet-18 on CIFAR-100 and Tiny-ImageNet. The results are presented in Table 3, where we observe the following: ❶ LoR-VP with LP as the output transformation achieves the overall best performance across all methods, networks, and datasets. ❷ Even when using the same output transformations as ILM-VP and AutoVP, LoR-VP consistently outperforms these methods, further demonstrating the superiority of our visual prompt design.

**What is the optimal rank in LoR-VP?** To provide deeper insights into the optimal rank selection in LoR-VP, we conduct experiments with various configurations: LoR-VP, LoR-VP combined with ILM as the output transformation named as LoR-VP w. ILM, and LoR-VP combined with FM as the output transformation named as LoR-VP w. FM. These experiments are performed using the ImageNet-21K-P pre-trained ViT-B/16-P on Tiny-ImageNet and the ImageNet-21K pre-trained ViT-B/32 (fine-tuned on ImageNet-1K) on CIFAR-100. The results are presented in Figure 6, from which we can derive the following observations: ❶ For output transformations such as LP and FM, the optimal rank is 4; increasing the rank beyond 4 does not yield any further performance improvements. ❷ When LoR-VP is combined with ILM, the optimal rank is around 16. A plausible explanation is that ILM lacks tunable parameters, so a higher rank is needed to enhance the expressive power of the visual prompts and achieve optimal performance.

Table 3: **The Impact of Output Transformation.** The performance comparison of utilizing FLM, ILM, and FM as the output transformation of LOR-VP and the baselines. LOR-VP achieves the overall best performance among all output transformation methods, networks, and datasets.

| Dataset | Method | Output Transformation | Network | | | |
|---|---|---|---|---|---|---|
| | | | Swin-B | ViT-B/16-P | ViT-B/32 | ResNet-18 |
| Tiny-ImageNet | ILM-VP[CVPR23] | ILM | 56.28 | 26.17 | 32.58 | 14.13 |
| | AutoVP[ICLR24] | FM | 84.81 | 81.42 | 82.43 | 59.68 |
| | LP | LP | 86.54 | 82.75 | 83.95 | 65.17 |
| | LOR-VP w. FLM | FLM | 82.15 | 41.76 | 82.89 | 57.45 |
| | LOR-VP w. ILM | ILM | 84.85 | 43.50 | 84.86 | 62.20 |
| | LOR-VP w. FM | FM | 85.59 | 83.15 | **86.03** | 65.63 |
| | LOR-VP | LP | **88.28** | **84.40** | 85.85 | **68.28** |
| CIFAR100 | ILM-VP[CVPR23] | ILM | 65.78 | 41.49 | 40.10 | 25.36 |
| | AutoVP[ICLR24] | FM | 86.83 | 88.58 | 85.96 | 63.77 |
| | LP | LP | 87.37 | 88.90 | 86.21 | 67.06 |
| | LOR-VP w. FLM | FLM | 74.08 | 46.35 | 72.81 | 34.58 |
| | LOR-VP w. ILM | ILM | 77.22 | 48.53 | 78.23 | 39.07 |
| | LOR-VP w. FM | FM | 86.25 | 89.10 | 88.48 | 68.64 |
| | LOR-VP | LP | **90.42** | **89.69** | **88.65** | **69.88** |

**Ablation of Components in LOR-VP.**  We perform ablation studies on the two key components of LOR-VP: the low-rank VP design and the linear probing output transformation. For experiments without label mapping, we apply FLM prior to training and keep this mapping sequence fixed during visual prompt training to ensure valid results. We conduct these experiments using the ImageNet-21K-P pre-trained ViT-B/16-P, ImageNet-21K pre-trained Swin-B, and ImageNet-1K pre-trained ResNet-18 on Tiny-ImageNet. The results, shown in Table 5, reveal the following: ❶ LOR-VP achieves the highest performance when both the low-rank VP and the output transformation are employed, demonstrating the effectiveness of these components in LOR-VP. ❷ Our VP design improves model performance, regardless of whether a fixed mapping sequence or linear probing is used.

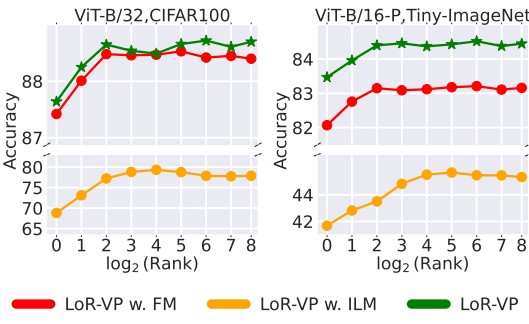

Figure 6: The Impact of Rank in LOR-VP.

Table 5: Ablation of Components in LOR-VP. The results of using ImageNet-21K-P pre-trained ViT-B/16-P, ImageNet-21K pre-trained Swin-B, and ImageNet-1K pre-train ResNet-18 on Tiny-ImageNet.

| Components | | Network | | |
|---|---|---|---|---|
| VP | Output Trans. | ViT-B/16-P | Swin-B | ResNet-18 |
| ✗ | ✗ | 36.09 | 80.61 | 40.79 |
| ✓ | ✗ | 41.76 | 82.15 | 57.45 |
| ✗ | ✓ | 82.75 | 86.54 | 65.17 |
| ✓ | ✓ | 84.40 | 88.28 | 68.28 |

## 6  CONCLUSION

Visual prompting has emerged as a powerful technique for adapting pre-trained models to specific tasks through parameter-efficient tuning. Traditional methods, however, often restrict the interaction between visual prompts and the original image to a limited number of patches, overlooking the potential benefits of shared information across different patches. Addressing these shortcomings, our study introduces a novel approach, termed Low-Rank Matrix Multiplication for Visual Prompting (LOR-VP), which facilitates both shared and patch-specific information dissemination throughout the image. Extensive experiments over seven networks and eight datasets consistently demonstrate the effectiveness and efficiency of our method.

## 7 ACKNOWLEDGEMENTS

This research is partially funded by research grants to Metaxas from NSF: 2310966, 2235405, 2212301, 2003874, 1951890, AFOSR 23RT0630, and NIH 2R01HL127661.

## 8 REPRODUCIBILITY STATEMENT

The authors have made an extensive effort to ensure the reproducibility of the results presented in the paper. *First*, the details of the experimental settings are provided Section 5.1 and in the Appendix A. This paper investigates eight datasets, and the details about each dataset are described in Table 6. The evaluation metrics are also clearly introduced in Section 5.1. *Second*, the baseline methods' implementation particulars are elucidated in Section 5.1. Simultaneously, the implementation details of our method, LOR-VP, are included in Section 5.1 and Appendix A. *Third*, the codes are included in the supplementary material for further reference.

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

# A  IMPLEMENTATION DETAILS

Table 6: Dataset Information.

| Dataset | Original Resolution | # Training Set Images | # Test Set Images | # Classes |
|---|---|---|---|---|
| ImageNet-21K (Deng et al., 2009) | Varies | 14M | - | 21,843 |
| ImageNet-21K-P (Ridnik et al., 2021) | $224 \times 224$ | 12M | 0.6M | 11,221 |
| ImageNet-1K (Deng et al., 2009) | Varies | 1.3M | 50K | 1,000 |
| ImageNet-R (Hendrycks et al., 2021a) | Varies | - | 30K | 200 |
| ImageNet-Sketch (Wang et al., 2019) | Varies | - | 50K | 1,000 |
| ImageNet-A (Hendrycks et al., 2021a) | Varies | - | 7.5K | 1,000 |
| ImageNet-V2 (Recht et al., 2019) | Varies | - | 10K | 1,000 |
| Tiny-ImageNet (Le & Yang, 2015) | $64 \times 64$ | 100K | 10K | 200 |
| CIFAR100 (Krizhevsky et al., 2009) | $32 \times 32$ | 50K | 10K | 100 |
| CIFAR10 (Krizhevsky et al., 2009) | $32 \times 32$ | 50K | 10K | 10 |

Table 7: Network Information.

| Network | Pre-trained Dataset | # Model Params | Resolution | Classifier Head Input Features | Classifier Head Output Features |
|---|---|---|---|---|---|
| ResNet-18 (He et al., 2016) | ImageNet-1K | 12M | $224 \times 224$ | 512 | 1,000 |
| ResNet-50 (He et al., 2016) | ImageNet-1K | 26M | $224 \times 224$ | 2,048 | 1,000 |
| ResNet-50-P (He et al., 2016) | ImageNet-21K-P | 46M | $224 \times 224$ | 2,048 | 11,221 |
| ViT-B/16-P (Dosovitskiy et al., 2020) | ImageNet-21K-P | 94M | $224 \times 224$ | 768 | 11,221 |
| ViT-B/16 (Dosovitskiy et al., 2020) | ImageNet-21K, ImageNet-1K | 87M | $224 \times 224$ | 768 | 1,000 |
| ViT-B/32 (Dosovitskiy et al., 2020) | ImageNet-21K, ImageNet-1K | 88M | $224 \times 224$ | 768 | 1,000 |
| Swin-B (Liu et al., 2021) | ImageNet-21K | 109M | $224 \times 224$ | 1,024 | 21,841 |
| CLIP (Radford et al., 2021) | WebImageText | 86M | $224 \times 224$ | 512 | - |

Table 8: Implementation Details.

| Network | Pre-trained Data | Downstream Data | Resolution | Optimizer | LR | Label Mapping | LoR-VP Rank | Epochs | Batch Size |
|---|---|---|---|---|---|---|---|---|---|
| ResNet-18 | ImageNet-1K | CIFAR100 | $224 \times 224$ | SGD | 0.02 | Linear Probing | 4 | 20 | 256 |
| ResNet-50 | ImageNet-1K | CIFAR100 | $224 \times 224$ | SGD | 0.02 | Linear Probing | 4 | 20 | 256 |
| ViT-B/32 | ImageNet-21K, ImageNet-1K | CIFAR100 | $224 \times 224$ | SGD | 0.02 | Linear Probing | 4 | 20 | 256 |
| ResNet-50-P | ImageNet-21K-P | Tiny-ImageNet | $224 \times 224$ | SGD | 0.02 | Linear Probing | 4 | 20 | 256 |
| ViT-B/16-P | ImageNet-21K-P | Tiny-ImageNet | $224 \times 224$ | SGD | 0.02 | Linear Probing | 4 | 20 | 256 |
| Swin-B | ImageNet-21K | Tiny-ImageNet | $224 \times 224$ | SGD | 0.02 | Linear Probing | 4 | 20 | 256 |
| CLIP | WebImageText | Tiny-ImageNet | $224 \times 224$ | SGD | 40 | Linear Probing | 4 | 20 | 256 |
| Swin-B | ImageNet-21K | ImageNet-1K | $224 \times 224$ | SGD | 0.01 | Linear Probing | 4 | 10 | 256 |

# B  ADDITIONAL INVESTIGATION

**Visual Prompting in Object Detection and Semantic Segmentation.**  In this paper, we primarily focus on image classification tasks, following previous works such as AutoVP and ILM-VP. To further explore the applicability of LoR-VP to object detection and semantic segmentation tasks, we conduct experiments using YOLOv4 (Bochkovskiy et al., 2020) for detection and DeepLabv3+ (Chen et al., 2018) for segmentation. Both models use ImageNet-1K pre-trained ResNet-50 as the backbone. Hyperparameters such as the number of epochs and the rank in LoR-VP are kept consistent with those used in classification tasks.

For object detection, we train on the Pascal VOC 2012 and 2007 training sets and evaluate on the Pascal VOC 2007 test set, following the setup in He et al. (2020). The bounding box head is modified for output transformation, and a learning rate of 0.0001 is applied. For semantic segmentation, we train on the Pascal VOC 2012 training set and evaluate on its validation set, with the DeepLabv3+ head adapted for downstream segmentation and a learning rate of 0.01. The experimental results for detection are presented in Table 9, while the segmentation results are shown in Table 10. Our method, LoR-VP, outperforms AutoVP by nearly $4\%$ in $AP_{50}$ on VOC 2007 detection and by $1.1\%$ on VOC 2012 segmentation, demonstrating the effectiveness of LoR-VP for object detection and semantic segmentation tasks.

**Additional Investigation on Diverse Downstream Tasks.**  To assess the performance of LoR-VP across a broader range of classification tasks, including those involving natural and artificial

Table 9: Performance (AP, AP$_{50}$, and AP$_{75}$) for object detection using YOLOv4 with an ImageNet-1K pre-trained ResNet-50 backbone, evaluated on the Pascal VOC 2007 test set.

| Method | AP | AP$_{50}$ | AP$_{75}$ |
|--------|-----|-----|-----|
| LP | 42.87 | 75.25 | 47.74 |
| AutoVP | 41.72 | 73.07 | 44.85 |
| LOR-VP | **43.21** | **77.02** | **48.07** |

Table 10: Performance (mIOU) for semantic segmentation using DeepLabv3+ with an ImageNet-1K pre-trained ResNet-50 backbone, evaluated on Pascal VOC 2012 validation set.

| Method | mIOU |
|--------|------|
| LP | 67.82 |
| AutoVP | 67.42 |
| LOR-VP | **68.55** |

objects, scenes, and textures, we conduct experiments on ten downstream datasets. These experiments use ViT-B/32 pre-trained on ImageNet-21K and fine-tuned on ImageNet-1K, following the methodologies of AutoVP and ILM-VP, to further evaluate the generalization and robustness of our approach. Although LOR-VP primarily focuses on pixel-level visual prompt designs, we extend our comparison to include VPT-DEEP, as described by Jia et al. (2022), which modifies the transformer layers. This allows for a more comprehensive evaluation against additional baselines. The experimental results, presented in Table 11, show that LOR-VP achieves superior average performance across the ten datasets compared to VPT and AutoVP. Specifically, LOR-VP improves performance by $1.4\%$ over AutoVP and $1.3\%$ over VPT, further demonstrating its effectiveness in diverse scenarios and against a wider range of baselines.

Table 11: Comparison of accuracy between LOR-VP and four baseline methods using ViT-B/32 pre-trained on ImageNet-21K and fine-tuned on ImageNet-1K across ten datasets.

| Method | Tiny-ImageNet | EuroSAT | OxfordPets | Food101 | DTD | Flowers102 | CIFAR10 | CIFAR100 | SVHN | GTSRB | Average |
|--------|---------------|---------|------------|---------|-----|------------|---------|----------|------|-------|---------|
| LP | 83.95 | 95.67 | 91.90 | 82.18 | 69.83 | 97.98 | 96.51 | 86.21 | 83.09 | 83.21 | 87.05 |
| VPT [ECCV22] | 83.54 | 95.90 | **92.27** | 82.29 | 72.11 | 98.47 | 96.02 | 86.22 | 81.48 | **88.29** | 87.66 |
| ILM-VP [CVPR23] | 32.58 | 88.12 | 78.92 | 48.24 | 42.65 | 64.27 | 85.27 | 40.10 | 80.81 | 67.88 | 62.88 |
| AutoVP [ICLR24] | 82.43 | **96.25** | 92.12 | 82.86 | 70.81 | 98.42 | 95.45 | 85.96 | 85.24 | 86.39 | 87.59 |
| LOR-VP | **85.85** | **96.25** | 92.18 | **83.51** | **72.49** | **98.58** | **97.52** | **88.65** | **86.31** | 88.07 | **88.94** |

Table 12: Additional comparison of LOR-VP and AutoVP using LP and FM as output transformations, respectively.

| Dataset | Method | Output Transformation | Network | | | |
|---------|--------|----------------------|---------|------|------|------|
| | | | Swin-B | ViT-B/16-P | ViT-B/32 | ResNet-18 |
| Tiny-ImageNet | LP | LP | 86.54 | 82.75 | 83.95 | 65.17 |
| | AutoVP[ICLR24] | FM | 84.81 | 81.42 | 82.43 | 59.68 |
| | AutoVP w. LP[ICLR24] | LP | 86.45 | 82.92 | 83.31 | 65.58 |
| | LOR-VP w. FM | FM | 85.59 | 83.15 | **86.03** | 65.63 |
| | LOR-VP | LP | **88.28** | **84.40** | 85.85 | **68.28** |
| CIFAR100 | LP | LP | 87.37 | 88.90 | 86.21 | 67.06 |
| | AutoVP[ICLR24] | FM | 86.83 | 88.58 | 85.96 | 63.77 |
| | AutoVP w. LP[ICLR24] | LP | 88.70 | 89.34 | 87.00 | 68.10 |
| | LOR-VP w. FM | FM | 86.25 | 89.10 | 88.48 | 68.64 |
| | LOR-VP | LP | **90.42** | **89.69** | **88.65** | **69.88** |

**Additional Investigation of Output Transformation.** Table 3 presents the impact of output transformations on LOR-VP, demonstrating that LOR-VP outperforms baseline methods, such as AutoVP and ILM-VP, when using the same output transformations. These results validate the effectiveness of our visual prompt designs. To further examine the influence of different output transformations and reinforce the superiority of our approach, we conduct additional ablation studies comparing LOR-VP and AutoVP using LP and FM as output transformations. The experiments are performed on the same architectures and datasets as those in Table 3. The results, shown in Table 12, indicate that LOR-VP consistently outperforms AutoVP with LP as the output transformation

across all models and datasets. Interestingly, AutoVP with LP achieves higher performance than LP alone on CIFAR100 but performs comparably on Tiny-ImageNet. This variation may stem from the scaling factors employed in AutoVP, which likely affect visual prompting performance differently across datasets. Notably, LOR-VP adopts a fixed visual prompt size of $224 \times 224$, simplifying its design by avoiding the need to account for scaling size, further underscoring its simplicity and adaptability.

