# OpenReview forum: "LoR-VP: Low-Rank Visual Prompting for Efficient Vision Model Adaptation"
_ICLR.cc/2025/Conference — ICLR 2025 Poster_

### Official Review · Reviewer_PGL1 · 2024-10-21

**Soundness:** 2
**Presentation:** 3
**Contribution:** 3
**Rating:** 5
**Confidence:** 4

**Summary:**

This paper introduces Low-Rank Visual Prompting (LoR-VP), which uses low-rank matrix multiplication to generate visual prompt, enabling more efficient information sharing across image patches, taking the inductive biases among them into consideration. The authors conducted a preliminary study comparing current SOTA (AutoVP) with three new VP designs, proving that VP should combine the benefits of both patch-specific and shared visual prompting. Tested on several network architectures and datasets, the proposed approach reduces the number of tunable parameters by up to 18× and achieves up to 6× faster training times while improving performance by an average of 3.1%.

**Strengths:**

Organic integration of LoRA and VP: The originality and novelty of this paper lie in its clever integration of Low-Rank Adaptation (LoRA) with Visual Prompting (VP), two previously established concepts, to create a highly efficient and effective approach for adapting pre-trained vision models. While LoRA has been used to reduce the complexity of model fine-tuning, and Visual Prompting focuses on task-specific adaptation through input modification, the paper's innovation is in combining these methods in a seamless way that enhances both parameter efficiency and model performance. By introducing low-rank matrix multiplications into the visual prompting process, LOR-VP allows shared and patch-specific information across the entire image, significantly outperforming existing methods in both speed and accuracy.

Exemplary clarity of reasoning:  In the preliminary study, the authors provide a well-structured and logical explanation for their design choices. They clearly demonstrate why both patch-specific and shared information in visual prompts are necessary by highlighting the limitations of existing methods that treat patches independently or focus only on peripheral areas. Additionally, their decision not to scale down the image emphasizes the importance of retaining maximum information for accurate model adaptation. This clear, step-by-step reasoning effectively justifies the development of LOR-VP, ensuring the method addresses these shortcomings while optimizing performance.

**Weaknesses:**

The preliminary study lacks rigor in controlling variables, as the impact of image scaling is not isolated from the role of patch-specific information. While design 4 (Patch-Same) outperforms others, the study does not definitively clarify whether its success is due to shared prompting across patches or the fact that the image is not scaled, leaving ambiguity about the true cause of the performance improvement. This undermines the ability to attribute the gains solely to patch sharing.

In the methodology section, the paper lacks formal mathematical proof detailing how the information across rows and columns in the visual prompts is linked, which leaves the assumptions of inductive bias vague. Additionally, while the low-rank matrix approach is intended to capture shared information, it does not explicitly guarantee that the natural relationship between neighboring pixels is preserved, and the exact nature of the associations formed between pixels remains unclear, weakening the justification for its effectiveness.

The method's performance relies heavily on empirical results without offering strong theoretical guarantees about convergence, optimality, or robustness in different settings, which could limit its broader adoption in critical applications.

The ablation studies comparing different output transformations lack clarity in distinguishing the contributions of the output transformation versus the LOR-VP component. Simply showing that LOR-VP outperforms other methods under the same output transformation doesn't clarify whether the performance gains are primarily due to the low-rank adaptation (LoRA) or the output transformation itself. Additionally, there is a noticeable performance drop for ViT models when using ILM and FLM, while Swin models do not exhibit this behavior. The authors fail to investigate or explain this discrepancy, leaving a gap in understanding why certain architectures are more sensitive to specific label mapping methods. A more detailed analysis of these interactions and a clearer separation of the contributions from each component are needed for a more rigorous assessment.

The paper lacks a thorough analysis of failure cases or edge scenarios where the LOR-VP method may struggle, such as on noisy or adversarial images. While the authors conduct robust experiments across multiple datasets and architectures, there is no exploration of how the method performs under conditions that deviate from the standard datasets, like adversarial attacks or high levels of image noise. For example, in Section 5.3, where robustness is discussed, the evaluation focuses on out-of-distribution generalization but does not account for adversarial robustness or resilience to noise, which are critical factors for real-world deployment. Without this analysis, it is unclear how reliable or stable LOR-VP would be in challenging environments, potentially limiting its practical use in more demanding applications.

**Questions:**

Impact of Image Scaling vs. Patch-Specific Information: Can you provide additional experiments or controlled studies to isolate the impact of image scaling from patch-specific information, to clarify whether the performance gains in design 4 are due to shared prompting or the lack of image scaling?

Mathematical Proof for Row and Column Information Link: Could you include a more formal mathematical explanation or proof of how the information across rows and columns in the visual prompts is linked, ensuring that the inductive bias of neighboring pixels being more related than distant ones is preserved?

Theoretical Guarantees on Convergence and Robustness: Can you offer theoretical insights or guarantees about the convergence, optimality, or robustness of the LOR-VP method, to complement the empirical results and ensure its reliability in diverse settings?

Clarifying the Contributions of Output Transformation vs. LOR-VP: Could you conduct additional ablation studies to more clearly separate the impact of the output transformation from the LOR-VP component, particularly to clarify why ViT models show a significant performance drop with ILM and FLM while Swin models do not?

Analysis of Sensitivity to Label Mapping in Different Architectures: Can you explore why ViT models seem more sensitive to label mapping methods compared to Swin models, and provide a deeper investigation into the factors causing this discrepancy?

Handling Noisy or Adversarial Images: Can you include experiments testing LOR-VP's performance under adversarial attacks or in the presence of noise, to assess its robustness and ensure its reliability in more challenging or real-world scenarios?

---

> ### Author Response · Authors · 2024-11-23
> **Point-to-point Response (Part 1)**
>
> Thank you for recognizing that our method enables more efficient information sharing across image patches and significantly outperforms existing methods in both speed and accuracy. We appreciate your acknowledgment of our preliminary study, which provides a well-structured and logical explanation for our design choices. Our paper effectively highlights the limitations of existing methods that focus solely on peripheral areas. Our step-by-step reasoning provides a clear and robust justification for the development of LoR-VP. Below, we provide detailed responses to your questions.
>
> > W1&Q1:  While design 4 (Patch-Same) outperforms others, the study does not definitively clarify whether its success is due to shared prompting across patches or the fact that the image is not scaled. Can you provide additional experiments or controlled studies to isolate the impact of image scaling from patch-specific information, to clarify whether the performance gains in design 4 are due to shared prompting or the lack of image scaling?
>
> Thank you for the question. There seems to be a misunderstanding regarding the role of image scaling in our study. The performance differences observed are not related to scaling. In Figure 2 of our paper, we compare Patch-Same (Part 4 of Figure 1) with Patch-Free (Part 3 of Figure 1), both of which utilize a resized image resolution of $224 \times 224$. We adopt a resolution of $224 \times 224$ because Patch-Pad (Part 2 of Figure 1) demonstrates inferior performance, despite using patch-wise pad prompts. This approach disrupts the continuity of the original image by splitting it into discontiguous parts, leading to a loss of crucial information. In contrast, the performance advantage of Patch-Same over Patch-Free highlights the importance of shared prompting information across patches.
>
> > W2&Q2: the paper lacks formal mathematical proof detailing how the information across rows and columns in the visual prompts is linked, which leaves the assumptions of inductive bias vague. Additionally, while the low-rank matrix approach is intended to capture shared information, it does not explicitly guarantee that the natural relationship between neighboring pixels is preserved, and the exact nature of the associations formed between pixels remains unclear, weakening the justification for its effectiveness. Could you include a more formal mathematical explanation or proof of how the information across rows and columns in the visual prompts is linked, ensuring that the inductive bias of neighboring pixels being more related than distant ones is preserved?
>
> Thank you for the suggestion. Unlike Patch-Same (Part 4 of Figure 1), which introduces patch-wise shared prompt information by directly using the same visual prompt for each patch, LoR-VP incorporates shared row and column (and thus across-patch) prompt information through the use of two low-rank matrices,  $\mathbf{B}$  and  $\mathbf{A}$, as described in Section 4.1.
> Specifically,  $\mathbf{B}$ serves as a basis for column visual prompts, where the visual prompt in each column of the image is a linear combination of the columns in $\mathbf{B}$. This design introduces shared information among different columns of the visual prompts. Meanwhile, the coefficients for each column are represented by the columns in $\mathbf{A}$, thereby introducing column-specific prompt information. Similarly, by interpreting  $\mathbf{A}$ as a basis for row visual prompts and $\mathbf{B}$  as the coefficients of these row bases, we establish an analogous understanding of the inductive biases introduced across rows in the LoR-VP visual prompt design. This formulation ensures that the shared and specific prompt information is distributed across both rows and columns and, thus, patches. The empirical results presented in Figure 2 demonstrate the superiority of Patch-Same over Patch-Free and LoR-VP over Patch-Same, further validating the effectiveness of incorporating shared visual prompts across patches, rows, and columns.

---

> ### Author Response · Authors · 2024-11-23
> **Point-to-point Response (Part 2)**
>
> > W3&Q3: The method's performance relies heavily on empirical results without offering strong theoretical guarantees about convergence, optimality, or robustness in different settings, which could limit its broader adoption in critical applications. Can you offer theoretical insights or guarantees about the convergence, optimality, or robustness of the LOR-VP method, to complement the empirical results and ensure its reliability in diverse settings?
>
> Thank you for the suggestion. While LoR-VP extends low-rank adaptation methods to the pixel-level input space of deep neural networks, the underlying technique of low-rank adaptation is well-established and widely validated in both NLP and CV tasks, as demonstrated by LoRA [1, 2, 3]. Theoretical guarantees regarding the convergence and reliability of low-rank adaptation methods, including LoRA, are detailed in [4, 5], and we kindly refer the reviewer to these references for further insights. Additionally, the effectiveness and reliability of LoR-VP are comprehensively validated through the empirical results presented in the revised paper. These results span diverse tasks, including image classification, object detection, and semantic segmentation, demonstrating consistent performance improvements and robustness across different settings. These findings reinforce the applicability of LoR-VP as a reliable and effective method for various critical applications.
>
> > W4&Q4:  Simply showing that LOR-VP outperforms other methods under the same output transformation doesn't clarify whether the performance gains are primarily due to the low-rank adaptation (LoRA) or the output transformation itself. A more detailed analysis of these interactions and a clearer separation of the contributions from each component are needed for a more rigorous assessment.  Could you conduct additional ablation studies to more clearly separate the impact of the output transformation from the LOR-VP component?
>
> Thank you for the question. To clarify the contributions of different components in LoR-VP, we have provided detailed ablation results in Table 5 of the revised paper. These results investigate the impact of both the visual prompt design and the output transformation. Specifically, we observe that the output transformation in LoR-VP (LP) improves performance compared to frequency-based label mapping (FLM). Furthermore, our visual prompt design enhances performance under both FLM and LP, validating its effectiveness and its critical role in LoR-VP.
>
> To ensure a rigorous comparison, we control the output transformation of LoR-VP to match that of the baselines, as shown in Table 3 of our paper. The results demonstrate that our visual prompt design outperforms the designs used in baseline methods, further highlighting its superiority.
>
> Additionally, we conduct further experiments using FM and LP as output transformations for both LoR-VP and AutoVP, with the results presented in Table 12 of the revised paper. These experiments show that LoR-VP achieves better performance than the SOTA method AutoVP, regardless of whether LP or FM is used as the output transformation. We kindly refer the reviewer to these results for a more comprehensive understanding of the contributions of our visual prompt design to the overall performance of LoR-VP.

---

> ### Author Response · Authors · 2024-11-23
> **Point-to-point Response (Part 3)**
>
> > W4&Q5:  there is a noticeable performance drop for ViT models when using ILM and FLM, while Swin models do not exhibit this behavior. The authors fail to investigate or explain this discrepancy, leaving a gap in understanding why certain architectures are more sensitive to specific label mapping methods. Can you explore why ViT models seem more sensitive to label mapping methods compared to Swin models, and provide a deeper investigation into the factors causing this discrepancy?
>
> Thank you for the question. In the results presented in Table 3 of our paper, we observe that when using ViT-B/32 with ILM and FLM on Tiny-ImageNet and CIFAR-100, there is no significant performance drop compared to LoR-VP with LP and FM as the output transformations, which contradicts the reviewer’s observation. However, we acknowledge that when using ViT-B/16-P on Tiny-ImageNet and CIFAR-100, the performance of LoR-VP with FLM and ILM is lower than expected.
>
> Upon further investigation, we find that this discrepancy arises from the choice of optimizer. Specifically, LoR-VP with FLM and ILM use SGD in our experiments. When we switch to Adam as the optimizer while keeping all other hyperparameters unchanged, the performance improves significantly. For ViT-B/16-P on CIFAR-100, LoR-VP with ILM and FLM achieves performances of 71.36 and 67.68, respectively. Similarly, for Tiny-ImageNet, LoR-VP with ILM and FLM achieves performances of 72.65 and 69.42, respectively—marked improvements over the results reported in our paper.
>
> We admit that we did not extensively tune hyperparameters for each model and dataset combination, as LoR-VP consistently outperformed the baselines with default settings. We thank the reviewer for highlighting this issue and will include the updated results for LoR-VP with FLM and ILM using ViT-B/16-P in the final version of the paper.
>
> > W5&Q6: The paper lacks a thorough analysis of failure cases or edge scenarios where the LOR-VP method may struggle, such as on noisy or adversarial images. Can you include experiments testing LOR-VP's performance under adversarial attacks or in the presence of noise, to assess its robustness and ensure its reliability in more challenging or real-world scenarios?
>
> Thank you for the suggestion. While none of the current visual prompting baselines provide experiments involving adversarial attacks or noisy inputs, which makes direct benchmarking against them infeasible, we agree that this is an important direction for future research. Proposing a benchmark for such tasks, however, is beyond the scope of this paper.
> The reliability and robustness of LoR-VP are demonstrated through extensive experiments across different architectures, model sizes, and dataset scales, as presented in Figures 4 and 5 of the paper. Furthermore, we validate LoR-VP’s robustness to distributional shifts by evaluating its performance on four out-of-distribution datasets, with results shown in Table 1 of the paper.
> To further assess the effectiveness of LoR-VP in more complex scenarios, we evaluate its performance using ViT-B/32 on ten datasets encompassing natural and artificial objects, scenes, and textures. As detailed in Table 11 of the revised paper, LoR-VP consistently outperforms AutoVP and ILM-VP on these challenging datasets, demonstrating its robustness in diverse conditions. Additionally, results on object detection and semantic segmentation, shown in Tables 9 and 10, further highlight LoR-VP’s effectiveness across a range of tasks.
>
> *References:*
>
> [1] LoRA: Low-Rank Adaptation of Large Language Models. ICLR 2022
> [2] DoRA: Weight-Decomposed Low-Rank Adaptation. ICML 2024
> [3] Parameter-Efficient Fine-Tuning with Discrete Fourier Transform. ICML 2024
> [4] The expressive power of low-rank adaptation. ICLR 2024
> [5] GaLore: Memory-Efficient LLM Training by Gradient Low-Rank Projection. ICML 2024

---

> ### Comment · Reviewer_PGL1 · 2024-11-28
>
> I am very grateful to the author for explaining the questions I raised, especially the explanation that the performance of LoR-VP with FLM and ILM is lower than expected is due to the choice of optimizer, which cleared up a misunderstanding. The explanation on the performance differences Patch-same and Patch-Free is clarified. However, I am still uncertain about the reasoning behind how LoR-VP utilizes shared information between columns and rows, along with the Math behind it. The reason why \mathbf{B}  serves as a basis for column visual prompts, similarly why \mathbf{A}  serves as a basis for row visual prompts is not solid enough to persuade me with with what the author says about integrating horizontal and vertical features. My review remains marginally below the acceptance threshold.

---

### Official Review · Reviewer_pQoJ · 2024-10-28

**Soundness:** 3
**Presentation:** 3
**Contribution:** 3
**Rating:** 6
**Confidence:** 3

**Summary:**

This paper proposes the incorporation of low-rank matrix multiplication in visual prompting, resulting in improved performance compared to existing visual prompting methods, as well as enhanced training speed.

**Strengths:**

1. the paper is very well-written and easy to follow. The background part is very clear and detailed.
2. the method is simple yet effect. And the author gives good preliminary analysis that motivates the method, which makes a lot of sense to me.

**Weaknesses:**

1. Regarding the pad-based method, the authors state, "The VP parameters are restricted to interacting with the original image in a limited set of patches, leaving a substantial portion of the image unmodified." I find this assertion questionable. If the backbone is a ViT, the padded tokens will interact with the inner tokens through self-attention, potentially affecting the entire image.
2. The authors do not include a comparison or discussion of visual prompt tuning [1], which is a more prevalent method than those cited in the paper.
3. The dataset utilized in the out-of-distribution generation experiments is insufficient to demonstrate the robustness of the method. It employs several variants of ImageNet, which exhibit minimal domain gaps, and all tasks focus on general object recognition. I recommend using a benchmark similar to AutoVP [2], which includes fine-grained classification and domain-specific tasks for a more comprehensive evaluation.
4. The comparison between AutoVP and LP appears unusual, as LP seems to outperform AutoVP in most cases. This contradicts the conclusions drawn in the AutoVP paper. Additionally, the performance gap between LoR-VP and LP is minimal. What would be the outcome if AutoVP's output transformation were replaced with LP?
5. As a general PEFT method, the evaluation is limitated to image classification, without extension to other tasks such as segmentation, detection, or caption.

[1] Jia, Menglin, et al. "Visual prompt tuning." European Conference on Computer Vision. Cham: Springer Nature Switzerland, 2022.

[2] Tsao, Hsi-Ai, et al. "Autovp: An automated visual prompting framework and benchmark." arXiv preprint arXiv:2310.08381 (2023).

**Questions:**

Please see weakness

---

> ### Author Response · Authors · 2024-11-23
> **Point-to-point Response**
>
> Thank you for recognizing that our paper is well-written and easy to follow. We appreciate your acknowledgment of the clarity and detail in the background part, the simplicity and effectiveness of our method, and the quality of the preliminary analysis that motivates our approach. Below, we provide detailed responses to your questions.
>
> > W1: If the backbone is a ViT, the padded tokens will interact with the inner tokens through self-attention, potentially affecting the entire image.
>
> Thank you for the insightful point. We agree that in ViT architectures, the self-attention mechanism facilitates interactions between padded tokens and inner tokens. However, the propagation of information through self-attention depends on the network’s pre-trained attention patterns, which are learned without accounting for the presence of visual prompts (VPs). As a result, these patterns may not effectively amplify the task-specific signals introduced by the VPs in the periphery of the image. Our visual prompt designs address this limitation by directly modifying the pixel-level information of all patches. Additionally, we leverage the inductive biases present in the shared information across patches, allowing the pre-trained model to better adapt to downstream tasks.
>
> > W2&W3: The authors do not include a comparison or discussion of visual prompt tuning. Using a benchmark similar to AutoVP [2], which includes fine-grained classification and domain-specific tasks for a more comprehensive evaluation.
>
> Thank you for the comment. We follow prior works, such as ILM-VP and AutoVP, to primarily investigate pixel-level visual prompt designs as the baselines. To provide a more comprehensive evaluation, we include additional experiments using ImageNet-21K pre-trained and ImageNet-1K fine-tuned ViT-B/32 on ten datasets, encompassing natural and artificial objects, scenes, and textures, to further demonstrate the generalization ability of our method. We also extend our comparison to include VPT [1], which modifies the transformer layers. In these experiments, we compare LoR-VP against four baselines: LP, ILM-VP, AutoVP, and VPT, following the implementations outlined in the original papers. The results, presented in Table 11 of the revised paper, show that LoR-VP achieves the best average performance across the ten datasets, outperforming both VPT and AutoVP. These findings further validate the effectiveness of our method in diverse scenarios.
>
> > W4: The comparison between AutoVP and LP appears unusual, as LP seems to outperform AutoVP in most cases. This contradicts the conclusions drawn in the AutoVP paper. Additionally, the performance gap between LoR-VP and LP is minimal. What would be the outcome if AutoVP's output transformation were replaced with LP?
>
> Thank you for the comment. In our experiments, we observe that LP achieves better performance than what was reported in the AutoVP paper. To ensure the validity of our comparisons, we use our LP results instead of directly adopting the LP results from AutoVP. To address your concern, we conduct additional experiments to compare LoR-VP with AutoVP using linear probing (LP) as the output transformation. Following the same implementation as the output transformation investigation in our paper (Table 3). The results, presented in Table 12 of the revised paper, show that LoR-VP consistently outperforms AutoVP with LP across all models. This further demonstrates the effectiveness of our method when AutoVP employs LP as its output transformation.
>
> > W5:  the evaluation is limitated to image classification, without extension to other tasks such as segmentation, detection, or caption.
>
> Thank you for the suggestion. We follow the approach of previous works, such as AutoVP and ILM-VP, which primarily focus their investigations on image classification tasks. To address your concern, we conduct additional experiments to extend our evaluation to object detection and semantic segmentation tasks. We utilize YOLOv4 [3] for object detection and DeepLabv3+ [4] for semantic segmentation. Both models employ ImageNet-1K pre-trained ResNet-50 as the backbone. For object detection, we train on the Pascal VOC 2012 and 2007 training sets and evaluate on the Pascal VOC 2007 test set. The bounding box head is modified for output transformation. For semantic segmentation, we train on the Pascal VOC 2012 training set and evaluate on its validation set, adapting the DeepLabv3+ head for downstream segmentation. The experimental results are presented in Table 9 for detection and Table 10 for segmentation in the revised paper. LoR-VP demonstrates strong performance, outperforming AutoVP by nearly 4% in $\text{AP}_{50}$ on VOC 2007 detection and by 1.1% in mIOU on VOC 2012 segmentation.
>
> *References:*
>
> [3] Yolov4: Optimal speed and accuracy of object detection. ArXiv 2020
> [4] Encoder-decoder with atrous separable convolution for semantic image segmentation. ECCV 2018

---

> > ### Comment · Reviewer_pQoJ · 2024-11-24
> >
> > Thanks for the reply. My concerns have been well solved. This paper is well motivated and very well-written. The numerical experiments are comprehensive after revision. A potential problem is the improvements over LP are not substantial. I remain uncertain about the practical benefits of the proposed method in real applications. I'll keep my rating to borderline accept.

---

> > > ### Author Response · Authors · 2024-11-25
> > > **Thank you for the feedback!**
> > >
> > > Thank you for the thoughtful feedback and for recognizing that our revisions have addressed your concerns. We appreciate the reviewer’s acknowledgment of the paper’s strong motivation, clear writing, and comprehensive numerical experiments.
> > >
> > > Regarding the improvements over LP, the additional results presented in Table 11 using ViT-B/32 on ten downstream classification datasets show that LoR-VP achieves an average accuracy improvement of 1.9% over LP, with a notable 5% improvement on GTSRB. The practical benefits of our method extend beyond in-distribution accuracy gains, as LoR-VP offers superior generalization performance, training time efficiency, memory efficiency, and parameter efficiency compared to current SOTA visual prompting methods, as highlighted in Table 1 and Table 2. Furthermore, LoR-VP is highly versatile and applicable to diverse tasks, including classification, detection, and segmentation.
> > >
> > > Again, we sincerely thank you for the detailed review and constructive feedback. In the final version of our paper, we will incorporate the new results and additional discussions to further enhance the quality and impact of our work.

---

### Official Review · Reviewer_NrTN · 2024-11-04

**Soundness:** 3
**Presentation:** 3
**Contribution:** 3
**Rating:** 8
**Confidence:** 4

**Summary:**

The paper addresses the task of visual prompting for adapting pre-trained models to specific downstream tasks. The paper investigates the limitations of existing visual prompting techniques, which often based on padding. The paper also proposes a new visual prompting technique based on low-rank matrix multiplication.

**Strengths:**

- The paper identifies the limitations of existing visual prompting techniques, which often restrict interaction between visual prompts and the original image to a small set of patches.
- A novel visual prompt design based on low-rank matrix multiplication is proposed. This design allows for shared and patch-specific information across rows and columns of image pixels.
- The results are convincing and demonstrate performance and efficiency improvements. The authors include extensive experiments across seven network architectures and several datasets showing a performance improvement compared to state-of-the-art methods.
- The paper is well written and clear.

**Weaknesses:**

- A low-rank matrix multiplication is just one way of sharing information across patches. I am surprised that other approaches have not been tested.
- The conclusions in the paper are very superficial and do not offer a deeper insight into the experimental results and the strengths and weaknesses of the proposed approach.

**Questions:**

- Why are alternative approaches of sharing information across patches not explored?
- The paper uses linear probing to transform the labels from the source to the target domain. This is a very simple model and it is not clear why this is more appropriate than other transform models.
- If I understand correctly, all downstream tasks are image classification tasks. Would the approach be able to deal other downstream tasks, e.g. object detection or segmentation?

---

> ### Author Response · Authors · 2024-11-23
> **Point-to-point Response (Part 1)**
>
> Thank you for acknowledging our contributions, including the investigation of limitations in existing visual prompting techniques, the novelty of our method, the convincing results demonstrating performance and efficiency improvements, and the clarity and quality of our writing. We appreciate your feedback and have provided detailed responses to your questions below:
>
> > W1&Q1: Why are alternative approaches of sharing information across patches not explored?
>
> Thank you for the question. To address the idea of sharing information across patches, we explored the Patch-Same method, which enables shared prompting by initializing a single tunable patch of parameters and repeatedly applying it to all patches of the image (see Part 4 of Figure 1). While this approach facilitates shared visual prompting across patches, it imposes a strong constraint by forcing the shared information to be identical for all patches. As shown in Figure 2, the Patch-Same method achieves better performance than AutoVP on ViT-B/32 but yields comparable performance on ViT-B/16. This suggests that while Patch-Same encourages compact and shared visual prompts, it may overly constrain the learning process by limiting the diversity of the learned prompts, which could hinder its adaptability to more complex tasks.
>
> These findings motivated us to develop the Low-Rank matrix multiplication Visual Prompting (LoR-VP) method. LoR-VP not only addresses the limitations of Patch-Same by allowing more flexible parameter sharing but also aligns with the goals of parameter-efficient fine-tuning (PEFT). Specifically, LoR-VP minimizes parameter usage while maintaining ease of optimization and deployment, making it an efficient and practical choice for PEFT applications.
>
>
> > W2: The conclusions in the paper are very superficial and do not offer a deeper insight into the experimental results and the strengths and weaknesses of the proposed approach.
>
> Thank you for your feedback. To address your concern, we expand our discussion to provide deeper insights into the experimental results and the strengths and weaknesses of our proposed approach:
>
> In this paper, we present a preliminary study to investigate the limitations of the widely used pad prompting technique, which pads tunable visual prompts only at the periphery of the image (see Part 1 of Figure 1). Our investigation reveals two key findings:
>
> 1. **Preservation of original image information**: Utilizing a contiguous image in visual prompts is critical to maintain the integrity of the original image information.
> 2. **Balanced information sharing**: Effective visual prompts should combine shared information across patches while also accommodating patch-specific prompts.
>
> These findings are validated by the results presented in Figure 2. Furthermore, we conduct extensive experiments to highlight the strengths of our visual prompt design, including its **superior generalization performance, training time efficiency, memory efficiency, and parameter efficiency** compared to existing methods, as demonstrated in Figure 4/5 and Table 1/2 of the paper.
>
> Additionally, we delve into the impact of output transformations and rank selection in our LoR-VP method in Table 3 and Figure 6 of the revised paper. These investigations offer practical insights into selecting appropriate ranks for LoR-VP under varying output transformation scenarios, providing a deeper understanding of the method’s adaptability and effectiveness.
>
> > Q2: The paper uses linear probing to transform the labels from the source to the target domain. it is not clear why this is more appropriate than other transform models.
>
> Thank you for the question. In Section 4.2, we discussed our rationale for utilizing linear probing (LP) for output transformation. The primary motivation is its parameter efficiency compared to other methods, such as iterative label mapping (ILM) and full mapping (FM), particularly when working with large models and datasets.
> Using LP as the classifier head avoids adding additional MLP layers, which would otherwise alter its functionality and potentially degrade performance. By directly modifying the MLP as the classifier head, LP maintains simplicity and efficiency.
> Additionally, when scaling to large datasets and models, ILM and FM become computationally and memory-intensive. For example, when using ImageNet-21K pre-trained Swin-B and tuning on ImageNet-1K, ILM requires significant resources to compute and store the mapping sequences (e.g., a 21,841 × 1,000 matrix). In our experiments, even with an NVIDIA Quadro RTX8000 setup ($8 \times 48$GB GPUs), these requirements exceeded our available computational capacity. Similarly, AutoVP necessitates training a 21,841 × 1,000 fully connected layer for FM, which is significantly more resource-intensive than the 1,024 × 1,000 classifier used in LP.

---

> > ### Comment · Reviewer_NrTN · 2024-11-24
> >
> > Thank you for your responses. I appreciate your answers and am convinced that this is an interesting paper. I will keep my positive assessment of the paper.

---

> > > ### Author Response · Authors · 2024-11-25
> > > **Thank you for the feedback!**
> > >
> > > Thank you for the positive feedback and for recognizing the contributions of our paper. We sincerely appreciate your support and your acknowledgment of the paper’s merits. In the final version of our paper, we will incorporate the new results and discussions to further enhance the quality and impact of our work.

---

> ### Author Response · Authors · 2024-11-23
> **Point-to-point Response (Part 2)**
>
> > Q3: Would the approach be able to deal other downstream tasks, e.g. object detection or segmentation?
>
> Thank you for the question. We follow the approach of previous works, such as AutoVP and ILM-VP, which primarily focus their investigations on image classification tasks. To address your concern, we conduct additional experiments to extend our evaluation to object detection and semantic segmentation tasks. We utilize YOLOv4 [1] for object detection and DeepLabv3+ [2] for semantic segmentation. Both models employ ImageNet-1K pre-trained ResNet-50 as the backbone. We keep hyperparameters such as the number of epochs and the rank in LoR-VP consistent with those used in classification tasks.
> For object detection, we train on the Pascal VOC 2012 and 2007 training sets and evaluate on the Pascal VOC 2007 test set. The bounding box head is modified for output transformation, and we use a learning rate of 0.0001. For semantic segmentation, we train on the Pascal VOC 2012 training set and evaluate on its validation set, adapting the DeepLabv3+ head for downstream segmentation with a learning rate of 0.01.
> The experimental results are summarized in Table 9 for detection and Table 10 for segmentation in the revised paper. LoR-VP demonstrates strong performance, outperforming AutoVP by nearly 4\% in $\text{AP}_{50}$ on VOC 2007 detection and by 1.1% in mIOU on VOC 2012 segmentation. These results highlight the versatility and effectiveness of our method in extending to tasks beyond image classification, including object detection and semantic segmentation.
>
> *References:*
>
> [1] Yolov4: Optimal speed and accuracy of object detection. ArXiv 2020
> [2] Encoder-decoder with atrous separable convolution for semantic image segmentation. ECCV 2018

---

### Author Response · Authors · 2024-11-23
**Highlighted General Response**

We sincerely appreciate all reviewers’ time and efforts in reviewing our paper. We also thank all reviewers for the insightful and constructive suggestions, which helped improve our paper further. In addition to our point-by-point responses, we provide the following highlighted general responses.

**[GR1] Additional Investigations**

As mentioned by the reviewers, we conduct additional experiments to validate the effectiveness of our method on new tasks, datasets, and other circumstances. We list some of the experiments mentioned by multiple reviewers in the following:

- **Detection and Segmentation.**
To explore the applicability of LoR-VP to object detection and semantic segmentation tasks, we perform experiments using YOLOv4 for detection and DeepLabv3+ for segmentation. Both models utilize ImageNet-1K pre-trained ResNet-50 as the backbone.
For object detection, we train on the Pascal VOC 2012 and 2007 training sets and evaluate on the Pascal VOC 2007 test set. For semantic segmentation, we train on the Pascal VOC 2012 training set and evaluate on its validation set.
The experimental results for detection are presented in Table 9 of the revised paper, and the results for segmentation are shown in Table 10. LoR-VP achieves a 4% improvement in $\text{AP}_{50}$ over AutoVP on VOC 2007 detection and a 1.1% mIOU improvement on VOC 2012 segmentation, demonstrating its effectiveness on object detection and semantic segmentation tasks.


- **Diverse Downstream Classification.** To assess the performance of LoR-VP across a broader range of classification tasks, we conduct experiments on ten downstream datasets. These experiments use ViT-B/32 pre-trained on ImageNet-21K and fine-tuned on ImageNet-1K, to further evaluate the generalization and robustness of our approach. The experimental results, presented in Table 11 in the revised paper, show that LoR-VP achieves superior average performance across the ten datasets compared to the SOTA method AutoVP, further demonstrating its effectiveness in diverse scenarios.

**[GR2] Paper Revision**

The revision of the paper is updated, including all new experimental results and references. All changes are clearly marked in blue for the reviewers’ convenience. We remain committed to improving our paper to make meaningful contributions to the field.

We hope our pointwise responses below can clarify all reviewers’ confusion and alleviate all concerns. We thank all reviewers’ time again.

---

### Meta-Review · Area_Chair_ytD9 · 2024-12-22

**Metareview:**

The paper introduces Low-Rank Visual Prompting (LoR-VP), a novel approach that combines Low-Rank Adaptation (LoRA) with Visual Prompting (VP) to enhance model efficiency and performance in visual prompting tasks. The method shows promising results, reducing parameters and training times while improving accuracy. However, concerns have been raised regarding the theoretical foundations and the approach's effectiveness in broader applications. Despite these concerns, the final average rating leans towards acceptance.

**Additional Comments On Reviewer Discussion:**

In the initial review, the reviewers pointed out that the paper lacks rigorous control in experiments, leaving ambiguity about the causes of performance improvements. There is no formal mathematical proof supporting the inductive biases of the method, and the lack of theoretical guarantees on convergence and robustness limits its generalizability. The ablation studies are not clear enough to differentiate the contributions of LoR-VP and output transformations. Additionally, the paper does not explore failure cases such as adversarial robustness or noisy environments. While most of these concerns were addressed during the discussion phase, a few reviewers remain skeptical about the theoretical aspects and the broader significance of the approach. Based on the discussion, the meta-reviewer believes that the merits still outweigh the cons, and therefore recommends a borderline accept.

---

### Decision · Program_Chairs · 2025-01-22

Accept (Poster)